# Q-BENCH-VIDEO: BENCHMARKING THE VIDEO QUALITY UNDERSTANDING OF LMMs

## ABSTRACT

With the rising interest in research on Large Multi-modal Models (LMMs) for video understanding, many studies have emphasized general video comprehension capabilities, neglecting the **systematic exploration into video quality understanding**. To address this oversight, we introduce **Q-Bench-Video** in this paper, a new benchmark specifically designed to evaluate LMMs' proficiency in discerning video quality. **a)** To ensure video source diversity, **Q-Bench-Video** encompasses videos from natural scenes, AI-generated Content (AIGC), and Computer Graphics (CG). **b)** Building on the traditional multiple-choice questions format with the *Yes-or-No* and *What-How* categories, we include *Open-ended* questions to better evaluate complex scenarios. Additionally, we incorporate the **video pair quality comparison** question to enhance comprehensiveness. **c)** Beyond the traditional *Technical*, *Aesthetic*, and *Temporal* distortions, we have expanded our evaluation aspects to include the dimension of *AIGC* distortions, which addresses the increasing demand for video generation. Finally, we collect a total of 2,378 question-answer pairs and test them on 12 open-source & 5 proprietary LMMs. Our findings indicate that while LMMs have a foundational understanding of video quality, their performance remains incomplete and imprecise, with a notable discrepancy compared to human performance. Through **Q-Bench-Video**, we seek to catalyze community interest, stimulate further research, and unlock the untapped potential of LMMs to close the gap in video quality understanding.

## 1 INTRODUCTION

As the field of artificial intelligence (AI) continues to evolve, Large Multi-modal Models (LMMs) (Ye et al., 2024; Li et al., 2024a; Chen et al., 2024; Ke et al., 2023; Xu et al., 2024) are progressively utilized in high-level video understanding tasks. These models have shown remarkable capabilities in analyzing and interpreting the semantic content of videos, such as classifying objects, identifying actions, and recognizing events. However, **the aspect of video quality, which is vital for optimizing compression and transmission systems, enhancing viewer experience, and establishing standards for high-quality video generation, has received less attention.** Although numerous LMM video benchmarks (Fu et al., 2024; Fang et al., 2024; Wu et al., 2024a) have been developed to assess the semantic understanding of videos by LMMs comprehensively, benchmarks systematically targeting video quality are still lacking. Additionally, while semantic understanding is closely linked to high-level video information, the perception and understanding of low-level information are crucial in video quality (Chikkerur et al., 2011; Li et al., 2024b). Thus, current video benchmarks fail to adequately evaluate the video quality understanding capabilities of LMMs.

To address this gap, we introduce **Q-Bench-Video**, a novel benchmark specifically designed to systematically evaluate the video quality understanding of LMMs. As illustrated in Fig. 1, our benchmark encompasses a wide range of video content, including natural scenes, AI-generated Content (AIGC), and Computer Graphics (CG), ensuring diversity in video sources. In addition, to maintain a reasonable distribution of source video quality, we employ uniform sampling from video datasets that contain subjective quality annotations. This approach guarantees comprehensive coverage of the quality spectrum while avoiding imbalanced quality distributions. Moreover, we extend beyond traditional video evaluations by incorporating both multiple-choice questions (MCQs) and open-ended questions. This enables a more thorough analysis of LMMs' ability to discern video quality

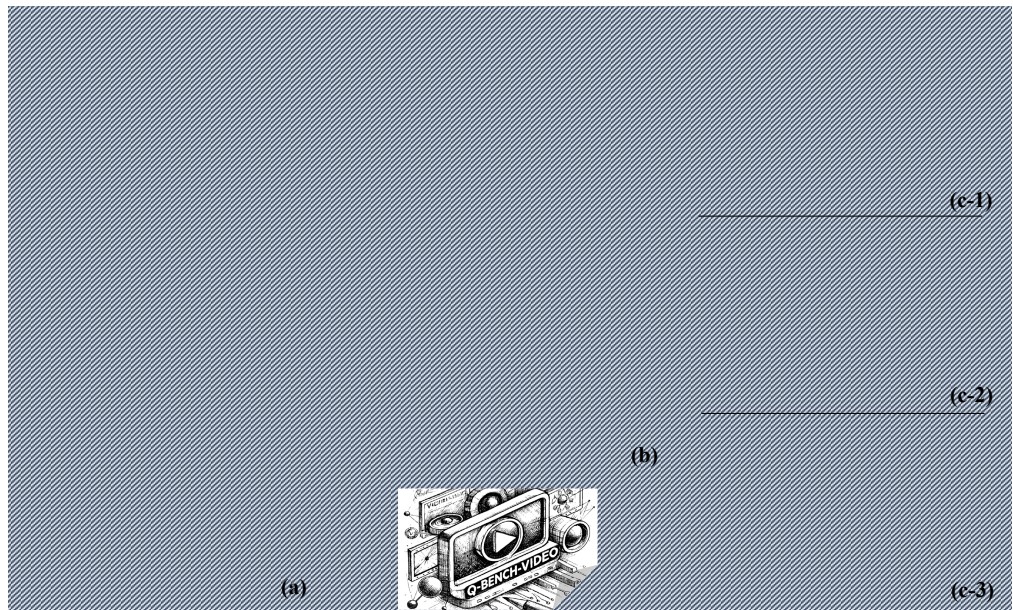

Figure 1: The construction overview of the proposed **Q-Bench-Video**. To ensure **diversity in video content**, we collect natural scenes, AIGC, and CG videos from video quality assessment datasets as depicted in (a). To achieve a balanced quality distribution among the sampled videos, we employ **uniform sampling for quality control**. As indicated in (c-1) and (c-2), we utilize three types of questions (*Yes-or-No, What-How, Open-ended*) and address a comprehensive range of quality concerns including *Technical*, *Aesthetic*, *Temporal*, and *AIGC* distortions. Additionally, we incorporate the video pairs comparison task to enhance the comprehensiveness of the benchmark.

across diverse scenarios. We further introduce a new evaluation dimension specifically tailored to assess distortions related to AIGC, which are increasingly prominent in video generation tasks. Recognizing the importance of quality comparison settings in real-world applications, such as camera parameter optimization and AIGC video generation, we further incorporate video pairs to facilitate quality comparison assessment as well. In total, we collect 1800 videos and annotated 2,378 question-answer pairs for validation, creating a robust framework for systematic evaluation.

Through the rigorous experiment, we demonstrate that while LMMs show promise in video quality assessment, their performance lags significantly behind human-level understanding. By offering a systematic and thorough evaluation of LMMs' video quality perception, **Q-Bench-Video** aims to foster research in this underexplored area and push the boundaries of LMM capabilities.

Our contributions can be summarized as follows:

- We introduce **Q-Bench-Video**, the first comprehensive benchmark explicitly designed to assess the video quality understanding capabilities of LMMs. This benchmark includes a diverse collection of source videos and ensures a balanced quality distribution, complemented by human-crafted question-answer annotations to enable thorough evaluation.

- Our evaluation framework spans four key quality dimensions: *Technical*, *Aesthetic*, *Temporal*, and *AIGC* distortions, which offers a holistic evaluation approach to video quality assessment. Uniquely, **Q-Bench-Video** enhances its utility by introducing the task of **video pairs comparison**, which sets it apart from existing video benchmarks.

- We conduct a comprehensive evaluation using both proprietary and open-source LMMs to measure their effectiveness in understanding video quality. The results expose notable deficiencies in current LMMs, while also shedding light on performance variations across different quality dimensions. These findings provide critical insights and suggest promising directions for future enhancements in the field of video quality understanding.

Table 1: Overview of the diverse video source datasets in the **Q-Bench-Video**. We consider various video content types, including **natural scenes, AIGC, and CG videos**. The term 'MOS' denotes that the videos are annotated via Mean Opinion Scores under ITU standards (itu, 2000). We have conducted uniform sampling based on their quality labels to ensure a **balanced quality distribution**.

| Video Type | Video Source Dataset | MOS | Quality Concerns | Sampled Size | Full Dataset Size |
|---|---|---|---|---|---|
| Natural (1000) | LSVQ (Ying et al., 2021) | ✓ | Spatial & Temporal | 600 | 39K |
| | MaxWell (Wu et al., 2023b) | ✓ | Spatial & Temporal & Aesthetic | 350 | 4.5K |
| | WaterlooSQoE-III (Duanmu et al., 2018) | ✓ | Quality-of-Experience | 20 | 450 |
| | WaterlooSQoE-IV (Duanmu et al., 2020) | ✓ | Quality-of-Experience | 30 | 1,350 |
| AIGC (600) | T2VQA-DB (Kou et al., 2024) | ✓ | Quality & Text Alignment | 200 | 10K |
| | VideoFeedback (He et al., 2024b) | ✗ | Quality & Text Alignment | 400 | 37.6K |
| CG (200) | LIVE-YT-Gaming (Yu et al., 2023) | ✓ | Visual Quality | 200 | 600 |

## 2 RELATED WORKS

### 2.1 LARGE MULTI-MODAL VIDEO MODELS AND BENCHMARKS

The rapid advancement of Large Multi-modal Models (LMMs) in recent years (Liu et al., 2023b;a; 2024a; Chen et al., 2024; Zhang et al., 2023a; Ye et al., 2023a) has showcased their remarkable perception and cognitive abilities across various multimodal benchmarks for images (Liu et al., 2023d; Wu et al., 2024b; Fu et al., 2023; Zhang et al., 2024d; Marino et al., 2019). As development progresses, the focus of visual analysis has gradually shifted from images to videos. Early efforts (Li et al., 2023a; Lin et al., 2023; Liu et al., 2023c; Xu et al., 2024) aimed at unlocking the video understanding potential of LMMs have yielded promising results. However, initial video-based benchmarks (Li et al., 2023b; Wang et al., 2023; Mangalam et al., 2023) typically concentrated on specific aspects of video comprehension, falling short of fully capturing the performance of these models due to limitations such as a lack of diversity in video types and inadequate coverage of temporal dynamics. In response, more recent video benchmarks (Fu et al., 2024; Fang et al., 2024) have moved toward a more comprehensive evaluation of LMMs. Nonetheless, these efforts primarily focus on high-level semantic understanding without systematic exploration of video quality.

### 2.2 VIDEO QUALITY ASSESSMENT

Video Quality Assessment (VQA) is a task aimed at quantifying video scores based on visual quality. Initially, early VQA methods employ hand-crafted features extracted from videos and regress the features into quality scores (Zheng et al., 2022; Vu et al., 2011; Li et al., 2018). With the emergence of deep neural networks, a shift occurr as numerous methods adopted deep learning techniques for VQA tasks (Li et al., 2019; Sun et al., 2022; Li et al., 2022; Wen et al., 2024). As the field progresses, newer methods begin incorporating considerations for both temporal dynamics and aesthetic qualities, leading to a more holistic approach to video quality analysis (Wu et al., 2022a; 2023a; Zhang et al., 2023b; Ahn & Lee, 2018; He et al., 2024a). Moreover, the evolution of large-scale models has further revolutionized VQA methodologies. Many recent approaches have redefined the traditional quality assessment process into a quality question-answering format (Wu et al., 2024c; Ge et al., 2024; Zhang et al., 2024b). This adaptation leverages the substantial prior knowledge embedded in large models to enhance the precision of quality quantification (Zhang et al., 2024c). Despite these technological advances, VQA still grapples with challenges in providing interpretable quality scores and deepening the understanding of how models perceive and analyze video quality.

## 3 BENCHMARK CONSTRUCTION

### 3.1 BENCHMARK PRINCIPLE

The **Q-Bench-Video** is designed based on three guiding principles: (**1**) **It encompasses a broad spectrum of video content**, including natural scenes, AIGC, and CG videos. (**2**) **It ensures a comprehensive and representative sampling process across a wide quality range**, enhancing the benchmark's overall effectiveness. (**3**) **It primarily focuses on the aspects of video quality that significantly influence the viewing experience**, including technical, aesthetic, temporal, and AIGC distortions. This significantly differs from other video benchmarks that prioritize semantic understanding. Additionally, the **video pair quality comparison** is integrated to address the challenges associated with comparing video quality. The construction can be overviewed in Fig. 1.

## 3.2 SOURCE VIDEOS COLLECTION

As shown in Table 1, the source videos are primarily gathered from video quality assessment datasets. We selected videos from these datasets for two main reasons: (**1**) These datasets have inherently considered the diversity of quality features during video selection; (**2**) These datasets possess quality annotations that adhere to ITU standards (itu, 2000) (with the exception of VideoFeedback), which allow us to accurately and authentically sample videos.

Our sampling method primarily employs a uniform approach, extracting videos evenly from each dataset based on the quality range. Moreover, considering the current popularity of AIGC and CG videos, we have also incorporated a selection of these video types. For a detailed description of the datasets and the sampling procedure, please refer to Appendix A and Appendix B.

## 3.3 BENCHMARK DESIGNS

In this section, we provide a detailed description of the design of **Q-Bench-Video**. In this benchmark, the meta-structure tuple $(V, Q, A, C)$ of each data item can be decomposed into several components: the video object $V$ (which can be a single video or a pair of videos), the video quality query $Q$, the set of possible answers $A$, and the correct answer $C$. The question samples are listed in Fig. 2.

### 3.3.1 QUESTION TYPES

**Yes-or-No Questions.** The basic *Yes-or-No* questions are designed to prompt LMMs to make binary judgments on video quality queries, typically limited to the answers *Yes* or *No*. To address the potential bias in LMMs that may skew towards *yes* responses, we employ a rigorous annotation process. This process ensures that the distribution of correct answers, either *Yes* or *No*, remains balanced at about 50%/50% ratio (see Appendix D). This balanced approach allows for a more accurate assessment of LMMs' performance on *Yes-or-No* questions.

**What-How Questions.** The *What-How* questions are commonly utilized in benchmarks for LMMs. The *What* questions focus on identifying specific distortions (*e.g., What is the most apparent distortion in this video?*). On the other hand, the *How* questions are employed to distinguish the finer details of distortion levels (*e.g., How is the overall clarity of this video?*). Including both *What* and *How* questions allows **Q-Bench-Video** to thoroughly and meticulously evaluate LMMs' ability on identifying video distortions and evaluating the distortion levels.

**Open-ended Questions.** It's important to note that the two types of questions previously mentioned require LMMs to select the correct answer from a predefined set. However, in many real-world scenarios, *Open-ended Questions*, which do not restrict responses to a predefined set, are often more necessary and challenging for LMMs (*e.g., What are the possible factors that lead to the low clarity of this video? Please list and explain.*). By adopting this form of questioning, we can better assess an LMM's ability to perceive video quality in real-world conditions.

### 3.3.2 QUALITY CONCERNS

It's important to recognize that video quality can be influenced by multiple factors on some occasions. Therefore, a query tuple $(V, Q, A, C)$ **does not need to be restricted to a single concern. It can address multiple concerns simultaneously.** For instance, the question *Is this video clear and well-composed?* can be seen as evaluating both technical and aesthetic quality understanding.

**Technical Distortions.** Technical distortions refer to the *low-level degradation* in video quality that arises from the limitations of recording, compression, and transmission (Su et al., 2021; Ying et al., 2021). These distortions often include artifacts such as *blurring, noise, compression artifact, exposure, etc.*, which are directly tied to the technical processes used in video production and delivery.

**Aesthetic Distortions.** Aesthetic distortions involve deviations from the *intended visual style, artistic design, or creative intent that negatively affect the viewer's perception* of the video (Wu et al., 2022b; Huang et al., 2024). These distortions can include aspects such as *confusing color, poor composition, lighting inconsistencies, or distracting elements* that reduce the overall aesthetic appeal. Unlike technical distortions, aesthetic distortions are subjective and might be affected by viewer preference, cultural context, or artistic norms.

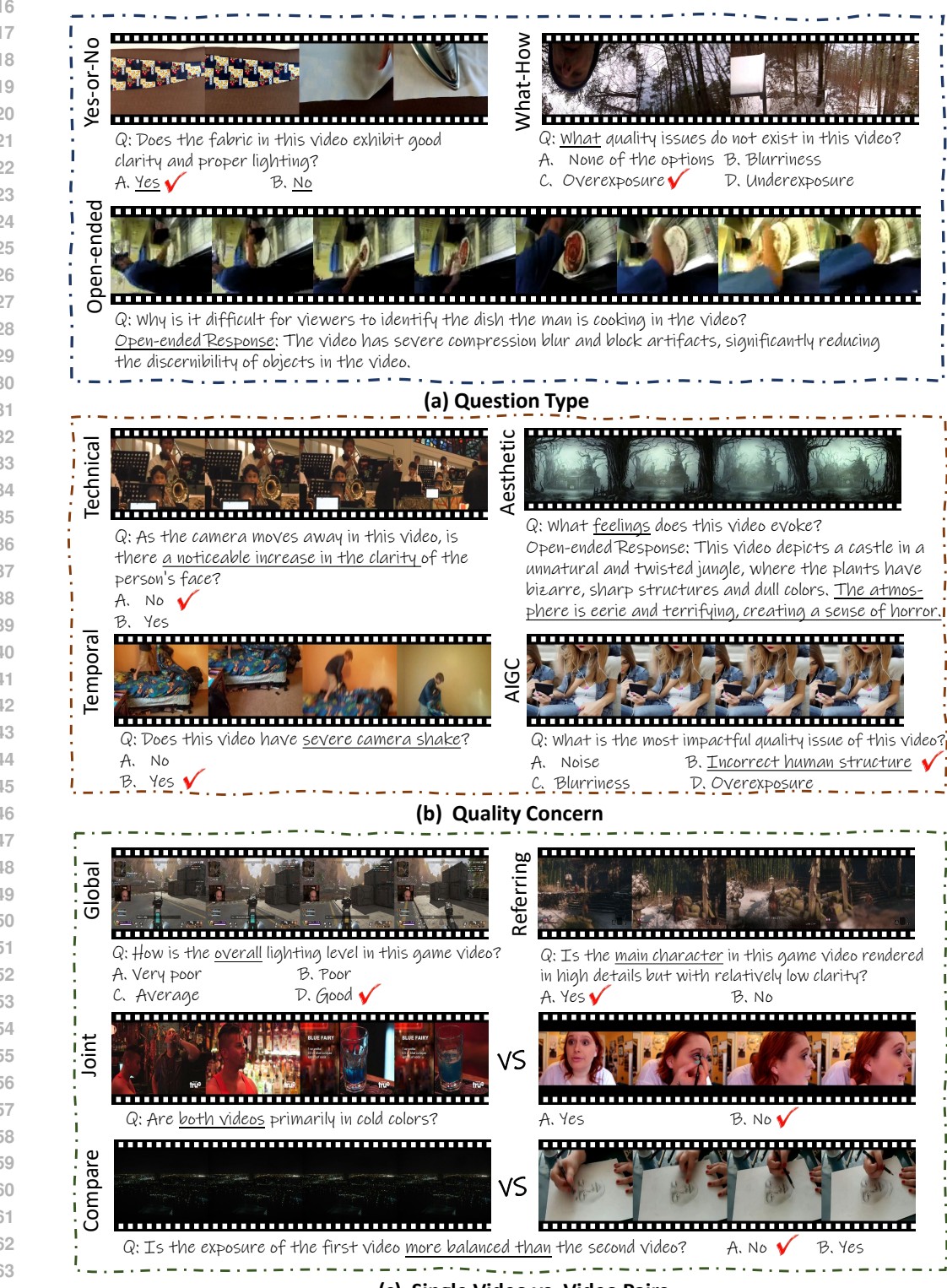

Figure 2: The visualization samples from **Q-Bench-Video**, with the question-answer content most representative of each subcategory being underlined. It is important to note that, regarding quality concerns, a single question-answer annotation may not only focus on one distortion dimension. Therefore, the distortion visualization examples shown in (b) primarily highlight instances that are most closely aligned with the mentioned distortion types.

**Temporal Distortions.** Temporal distortions are related to the *degradation of visual quality over time*, impacting the fluidity and consistency of the video (Seshadrinathan & Bovik, 2009). These distortions manifest as issues like *screen shake, flickering, motion inconsistency, frame drops, and stuttering* that result from unstable shooting devices, dramatically changing lighting conditions, and unstable bitrate environments (Wu et al., 2023b). Such disruptions hinder the viewer's natural perception of the video, leading to a disjointed and unpleasant viewing experience.

**AIGC Distortions.** AIGC distortions pertain to *imperfections and unnaturalness specifically arising from the generation of video content through AI models* (Liu et al., 2024b; Zhang et al., 2024c). These distortions may include *unnatural textures, inconsistent lighting, uncanny facial features, or unrealistic object behavior* that result from limitations or biases in the training data, model architecture, or generative process. These distortions are unique to AI-generated content and require specialized evaluation metrics that consider both the technical and perceptual quality aspects.

### 3.3.3 SINGLE VIDEOS & VIDEO PAIRS

Accurately comparing and jointly analyzing the quality of video pairs is sometimes more crucial than assessing the quality of a single video, especially in scenarios such as performance tests in video compression and quality control in video generation (In which it is more important to find out *Which video is better in visual quality and why?*) (Zhu et al., 2024; Wu et al., 2024d). Therefore, in **Q-Bench-Video**, we include video quality queries for both **single videos** and **video pairs**.

**Single Videos.** Queries related to single videos can primarily be categorized into two types: **a**) **Global perception**, which involves questions about the overall visual quality of the video, such as *How is the overall contrast of this video?* **b**) **Referring perception**, which focuses on the visual quality of specific elements within the video, like querying *What is the most apparent distortion when the player strikes the ball?* Through these approaches, we aim to comprehensively evaluate the LMMs' ability to perceive both the overall and localized aspects of video quality.

**Video Pairs.** Firstly, to ensure that the comparison of video pairs is clear and meaningful, **comparisons are only made between videos from the same source**, such as videos from natural sources being paired together while CG videos and AIGC videos are not being paired. There are mainly two types of video pair categories: **a**) **Joint analysis**, which involves understanding the shared quality features of the video pairs, for example, such as asking *Are both videos blurry?* **b**) **Comparative analysis**, which involves comparing the quality dimension across two videos, such as *How does the level of brightness in the first video compare to that in the second one?* It is important to note that we further categorize comparisons based on the difference in quality labels of the videos involved into **coarse-grain** (with relatively more significant visual quality differences) and **fine-grain** (with relatively minor visual quality differences) comparisons. More details can be found in Appendix J.

### 3.3.4 QUESTIONS & ANSWERS ANNOTATION

The annotation process of **Q-Bench-Video** is conducted in a well-controlled laboratory environment. A total of 8 experts are employed and trained to ensure the consistency of the annotations. The experts are required to watch the videos in their entirety before making annotations. Each annotated question-answer pair is then reviewed by at least three other experts to ensure its validity and accuracy. The annotation details and GUI visualization are presented in Appendix C.

### 3.4 BENCHMARK SETTING & EVALUATION

Unless specifically stated otherwise, for *Video LMMs* we typically analyze by uniformly sampling 16 frames from the video, while for *Image LMMs*, the sampling is reduced to 8 frames. For **Yes-or-No** and **What-How** questions, if the LMMs can accurately respond with the options, we directly record the accuracy of the responses as results. If the LMMs cannot provide option-based answers, we implement a GPT-assisted evaluation strategy to help judge the accuracy of the answers. For **Open-ended** questions, since the answers are open-ended and cannot be directly quantified for accuracy, we also employ the GPT-assisted evaluation strategy. This involves GPT scoring the responses based on their accuracy, completeness, and relevance compared to the annotated answer. Details about benchmark setting and evaluation can be found in the Appendix E and Appendix G.

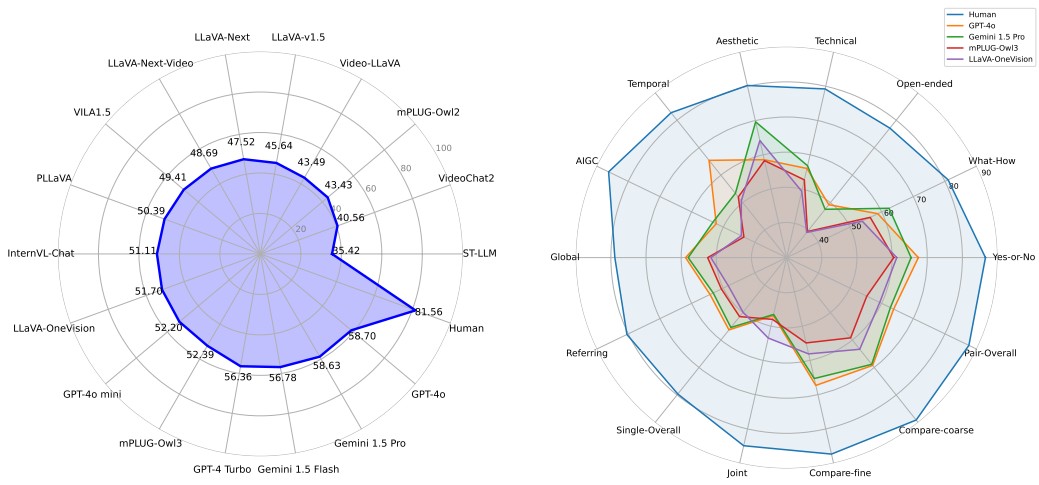

(a) Overall performance on **Q-Bench-Video**.  (b) Subcategory performance on **Q-Bench-Video**.

Figure 3: A concise summary of the LMMs' performance on **Q-Bench-Video**. (a) provides a comparison detailing the overall performance of humans and 17 selected LMMs, including both *proprietary* and *open-source* models. (b) illustrates a radar chart that outlines the performance of the **top-2** *proprietary* LMMs (GPT-4o & Gemini 1.5 Pro) and *open-source* LMMs (mPLUG-Owl3 & LLaVA-OneVision) across various subcategories within **Q-Bench-Video**.

## 4 RESULTS OF Q-BENCH-VIDEO

### 4.1 EXPERIMENTAL SETTING

**LMMs Participants.** A total of 17 LMMs (12 *Open-source LMMs* and 5 *Proprietary LMMs*) are included for validation, which includes **a**) 3 *Open-source Image LMMs*: LLaVA-Next (Liu et al., 2024a), LLaVA-v1.5 (Liu et al., 2023a), and mPLUG-Owl2 (Ye et al., 2023b); **b**) 9 *Open-source Video LMMs*: mPLUG-Owl3 (Ye et al., 2024), LLaVA-OneVision (Li et al., 2024a), InternVL-Chat (Chen et al., 2024), VILA1.5 (Ke et al., 2023), PLLaVA (Xu et al., 2024), LLaVA-Next-Video (Zhang et al., 2024a), ST-LLM (Liu et al., 2023c), Video-LLaVA (Lin et al., 2023), and VideoChat2 (Li et al., 2023a); **c**) 5 *Proprietary LMMs*: Gemini 1.5 Flash, Gemini 1.5 Pro (Team, 2024), GPT-4o mini, GPT-4o, and GPT-4 Turbo (Achiam et al., 2023).

**Subsets Split.** The **Q-Bench-Video** is divided into `test` (1,186 question-answer items) and `dev` (1,192 question-answer items) subsets. The correct answers will be released and proprietary for the `dev` and `test` subsets respectively. All discussions and analyses are based on the `test` subset. The details for the human performance on the `test` subset can be found in Appendix F.

### 4.2 FINDINGS

The overall performance and subcategory comparisons (human vs. top-performing LMMs) on **Q-Bench-Video** can be quickly glanced at Fig. 3. Detailed performance across the subcategories for each LMM are shown in Table 2 (**Question Types** & **Quality Concerns**) and Table 3 (**Single Videos vs. Video Pairs**) respectively. The organization of the findings is as follows:

**1) General Performance. Human>Proprietary LMMs>Open-source LMMs>Random guess.** From the performance results presented in Table 2, we observe that nearly all LMMs significantly outperform *random guess*, demonstrating their basic capability to understand video quality. Among the open-source LMMs, the recently released mPLUG-Owl3 achieves the highest *overall* performance at 52.39%, even slightly surpassing GPT-4o mini (52.20%), followed closely by LLaVA-OneVision (51.70%) and InternVL-Chat (51.11%). Image LMMs deliver moderate performance. Although they outperform some Video LMMs, the gap between them and the latest Video LMMs is still notable. Benefiting from larger training datasets and more parameters, proprietary LMMs (ex-

Table 2: Results on the `test` subset for the video quality perception ability of LMMs. The best performance is marked in **bold** and the second performance is underlined for *Open-source* and *Proprietary* LMMs respectively. The *Open-ended* questions are judged as 0.00% for **Random guess**.

| Sub-categories | Question Types | | | Quality Concerns | | | | |
|---|---|---|---|---|---|---|---|---|
| LMM *(LLM)* | *Yes-or-No↑* | *What-How↑* | *Open-ended↑* | *Tech.↑* | *Aes.↑* | *Temp.↑* | *AIGC↑* | *Overall↑* |
| **Random guess** | 50.00% | 25.00% | 0.00% | 23.70% | 23.46% | 25.83% | 21.69% | 25.67% |
| **Human** | 86.57% | 81.00% | 77.11% | 79.22% | 80.23% | 82.72% | 86.21% | 81.56% |
| *Open-source Image LMMs* | | | | | | | | |
| LLaVA-Next (*Mistral-7B*) | 62.83% | 45.14% | 33.69% | 46.38% | 57.86% | 47.84% | 48.46% | 47.52% |
| LLaVA-v1.5 (*Vicuna-v1.5-13B*) | 52.98% | 46.44% | 37.01% | 45.77% | 58.12% | 45.30% | 46.48% | 45.64% |
| mPLUG-Owl2 (*LLaMA2-7B*) | 59.19% | 39.07% | 31.19% | 42.07% | 52.38% | 41.71% | 39.37% | 43.43% |
| *Open-source Video LMMs* | | | | | | | | |
| mPLUG-Owl3 (*Qwen2-7B*) | 60.48% | **56.39%** | 39.48% | **52.68%** | 58.31% | **52.05%** | 43.49% | **52.39%** |
| LLaVA-OneVision (*Qwen2-7B*) | 61.34% | 53.88% | 39.15% | 49.35% | **64.15%** | 50.68% | 44.30% | 51.70% |
| InternVL-Chat (*Vicuna-7B*) | **66.02%** | 52.13% | 33.93% | 48.42% | 52.73% | 50.59% | 53.12% | 51.11% |
| VILA1.5 (*LLaMA3-8B*) | 61.95% | 46.00% | **39.60%** | 47.85% | 57.85% | 45.65% | 42.57% | 49.41% |
| PLLaVA (*Mistral-7B*) | 65.63% | 52.33% | 32.23% | 49.69% | 61.32% | 50.96% | 53.64% | 50.39% |
| LLaVA-Next-Video (*Mistral-7B*) | 61.34% | 45.95% | 38.10% | 49.03% | 60.94% | 46.97% | 49.40% | 48.69% |
| ST-LLM (*Vicuna-v1.1-7B*) | 44.63% | 28.50% | 32.78% | 34.99% | 46.11% | 34.28% | 34.02% | 35.42% |
| Video-LLaVA (*Vicuna-v1.5-7B*) | 64.67% | 40.79% | 29.11% | 43.25% | 54.04% | 42.38% | 42.76% | 43.49% |
| VideoChat2 (*Mistral-7B*) | 56.09% | 29.98% | 34.99% | 39.26% | 50.02% | 38.25% | 35.88% | 40.56% |
| *Proprietary LMMs* | | | | | | | | |
| **Gemini 1.5 Flash** | 65.48% | 56.79% | 47.51% | 54.11% | 66.58% | 53.51% | 50.22% | 56.78% |
| **Gemini 1.5 Pro** | 65.42% | **62.35%** | 47.57% | **56.80%** | **69.61%** | 53.38% | **53.26%** | 58.63% |
| **GPT-4o mini** | 62.95% | 50.93% | 42.10% | 49.38% | 60.90% | 48.43% | 41.71% | 52.20% |
| **GPT-4o** | **67.48%** | 58.79% | **49.25%** | 56.01% | 58.57% | **65.39%** | 52.22% | **58.70%** |
| **GPT-4 Turbo** | 66.93% | 58.33% | 40.15% | 54.23% | 66.23% | 54.00% | 52.04% | 56.36% |

cept GPT-4o mini) outperform all open-source models. However, even the best-performing model, GPT-4o, which achieved an *overall* performance of 58.70%, still lags behind human performance by 22.86%. This gap highlights that, despite the advancements in current state-of-the-art LMMs, there remains a significant need for improvement in video quality understanding ability.

**2) Question Types. *Open-ended* questions are more challenging for LMMs.** From Table 2, a discernible hierarchy in task difficulty for video quality assessment emerges for both humans and LMMs, arranged as follows: *Open-ended >What-How >Yes-or-No*. It is crucial to highlight that while humans exhibit a performance decline in *Open-ended* tasks by approximately 9.46% compared to *Yes-or-No* tasks, and about 3.89% compared to *What-How* tasks, these reductions are markedly less pronounced than those observed in LMMs for the *Open-ended* questions. This disparity underscores a significant proficiency gap between LMMs' capability in handling straightforward, closed-form questions and their effectiveness in navigating the complexities of real-world problem-solving, particularly in the context of video quality evaluation.

**3) Quality Concerns. LMMs exhibit unbalanced performance across different types of distortions.** From Table 2, it is evident that humans are particularly good at identifying *AIGC* distortions, while LMMs demonstrate stronger performance in detecting *Aesthetic* distortions. This distinction likely stems from the inherent sensitivity of humans to the conspicuous unnaturalness of *AIGC* distortions, which readily draws human attention. In contrast, *Aesthetic* distortions, which often involve high-level semantic nuances, align more closely with the training contexts of LMMs, enabling them to excel in this area. However, LMMs face challenges with *AIGC* distortions due to insufficient exposure to such anomalies during their pretraining phases, specific architectural constraints, and imperfections in the generation process. In the case of proprietary LMMs, their performance on *Technical* and *Temporal* distortions appear comparably consistent, indicating a uniform capability in recognizing these two types of distortions. Nonetheless, across all four subcategories, LMMs exhibit a notable performance disparity compared to humans, with varying degrees of accuracy among the different types of distortions. This variability highlights the need for significant enhancements in LMMs' abilities to accurately understand and interpret various distortion types.

Table 3: Results on the `test` subset for the video quality perception ability across single videos and video pairs of LMMs. The best performance is marked in **bold** and the second performance is underlined for *Open-source* and *Proprietary* LMMs respectively.

| Sub-categories | Single Videos | | | Video Pairs | | | |
|---|---|---|---|---|---|---|---|
| **LMM** (*LLM*) | *Global*↑ | *Referring*↑ | *Overall*↑ | *Joint*↑ | *Compare -fine*↑ | *Compare -coarse*↑ | *Overall*↑ |
| **Random guess** | 21.49% | 27.08% | 24.47% | 29.58% | 31.93% | 27.40% | 29.46% |
| **Human** | 78.87% | 80.43% | 79.65% | 84.90% | 87.34% | 89.11% | 87.56% |
| LLaVA-Next (*Mistral-7B*) | 51.33% | 50.20% | 50.73% | 38.03% | 48.00% | 42.48% | 43.46% |
| LLaVA-v1.5 (*Vicuna-v1.5-13B*) | 47.99% | 51.94% | 50.10% | 27.72% | 34.60% | 42.12% | 36.42% |
| mPLUG-Owl2 (*LLaMA2-7B*) | 46.86% | 43.51% | 45.07% | 51.49% | 37.10% | 40.28% | 43.69% |
| *Open-source Video LMMs* | | | | | | | |
| mPLUG-Owl3 (*Qwen2-7B*) | **52.46%** | 50.60% | 51.47% | 48.03% | 54.90% | 59.20% | 55.31% |
| LLaVA-OneVision (*Qwen2-7B*) | 51.56% | 48.43% | 49.89% | 53.48% | **58.10%** | **63.36%** | **59.41%** |
| InternVL-Chat (*Vicuna-7B*) | 51.15% | 51.86% | 51.52% | 48.85% | 51.10% | 49.20% | 49.79% |
| VILA1.5 (*LLaMA3-8B*) | 52.35% | 47.37% | 49.69% | 56.11% | 45.40% | 48.04% | 48.84% |
| PLLaVA (*Mistral-7B*) | 51.44% | **55.49%** | **53.60%** | 40.36% | 50.40% | 54.16% | 49.90% |
| LLaVA-Next-Video (*Mistral-7B*) | 51.33% | 50.20% | 50.73% | 38.03% | 48.00% | 42.48% | 43.46% |
| ST-LLM (*Vicuna-v1.1-7B*) | 36.54% | 36.49% | 36.51% | 28.03% | 36.80% | 32.08% | 32.87% |
| Video-LLaVA (*Vicuna-v1.5-7B*) | 45.46% | 44.67% | 45.04% | 49.36% | 42.00% | 43.00% | 44.01% |
| VideoChat2 (*Mistral-7B*) | 43.52% | 38.27% | 40.72% | **57.23%** | 44.40% | 41.64% | 45.93% |
| *Proprietary LMMs* | | | | | | | |
| **Gemini 1.5 Flash** | 58.00% | 53.18% | 55.43% | 46.59% | 65.30% | 68.84% | 62.77% |
| **Gemini 1.5 Pro** | 52.36% | **61.41%** | **57.19%** | 45.43% | 65.30% | **72.00%** | 63.55% |
| **GPT-4o mini** | 52.67% | 48.96% | 50.69% | 44.00% | 60.50% | 63.88% | 58.02% |
| **GPT-4o** | **58.75%** | 54.18% | 56.31% | **46.93%** | **67.30%** | 69.24% | **63.80%** |
| **GPT-4 Turbo** | 57.36% | 52.80% | 54.93% | 46.13% | 62.50% | 64.80% | 59.84% |

**4) Single Videos vs. Video Pairs. LMMs demonstrate superior capabilities in comparing video quality.** From Table 3, we observe that for single videos, LMMs achieve similar performance in *Global* and *Referring* quality perception (except for Gemini 1.5 Pro), without any significant trend of the performance for one subcategory over the other. This suggests that LMMs have comparable abilities in perceiving both *Global* video quality and *Referring* video quality. In terms of comparison, however, LMMs clearly outperform their performance on single video analysis and joint analysis. Notably, LMMs perform significantly better in the *Compare-coarse* subcategory, where video pairs have more pronounced quality differences, than in the *Compare-fine* subcategory. This highlights that LMMs are more adept at comparing video quality than analyzing the quality of single videos. This advantage in comparative assessment can be attributed to the inherent clarity in pairwise comparisons, which provide explicit contrasts, as opposed to the more ambiguous nature of evaluating a single video. Both humans and LMMs exhibit enhanced performance in comparative tasks. Although there is still a significant accuracy gap between LMMs and humans, LMMs show promising potential as effective tools for comparing video quality.

## 5 CONCLUSION

In this paper, we introduce **Q-Bench-Video**, the first comprehensive benchmark explicitly designed to evaluate Large Multi-modal Models' (LMMs) understanding of video quality. Our benchmark includes a diverse range of video types, questions that challenge multiple aspects of video quality, and a holistic evaluation framework encompassing *Technical*, *Aesthetic*, *Temporal*, and *AIGC* distortions. Through extensive experimentation with 17 *open-source* and *proprietary* LMMs, we find that while LMMs show promise in discerning video quality, their performance remains significantly below human-level understanding, especially when addressing *Open-ended* questions and *AIGC-specific* distortions. These findings highlight the current limitations of LMMs in video quality perception and underscore the need for further advancements in this area. By offering **Q-Bench-Video**, we aim to stimulate future research and drive improvements in the field, ultimately bridging the gap between LMM and human video quality assessment capabilities.

## 6 ETHICS STATEMENT

This submission fully complies with the ethical standards outlined by ICLR 2025. In particular, we adhere to ICLR's guidelines for responsible AI development, ensuring that our research does not contribute to harm, bias, or discrimination. All data used in this study is sourced from publicly available, ethically curated datasets, and our methodologies have been designed to promote fairness, accountability, and transparency in the evaluation of video quality.

Given the nature of evaluating video quality using large multi-modal models (LMMs), we have taken careful measures to ensure that the methodologies proposed in this study are applied in a way that promotes fair use and contributes positively to the field. We explicitly avoid the development of tools or systems that could be misused for deceptive or malicious purposes, such as content manipulation or exploitation. Our benchmark aims to support the responsible advancement of video quality assessment, which is critical for improving visual media technologies. We acknowledge the inherent risks of bias and fairness in the datasets used, particularly with AI-generated content (AIGC) and human evaluation. In this regard, we have applied uniform sampling methods across video datasets and employed a diverse set of human annotators to minimize subjective bias and ensure balanced quality distribution. The annotation processes were carefully designed and reviewed by multiple experts to ensure consistency and fairness across different video content types.

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

## A  SOURCE VIDEO DATASET INTRODUCTION

In this section, we briefly introduce the video quality assessment (VQA) datasets as follows:

- **LSVQ** (Ying et al., 2021): The LSVQ dataset is currently the largest VQA dataset, comprising over 39,000 real-world videos and 5.5 million human perceptual quality annotations. It primarily focuses on both *spatial and temporal aspects of technical visual quality*.

- **MaxWell** (Wu et al., 2023b): The MaxWell dataset presents a comprehensive subjective study, gathering over two million human opinions on 13 distinct quality factors across 4,543 in-the-wild natural scene videos. These quality factors include technical aspects (*sharpness, focus, noise, motion blur, flicker, exposure, compression artifacts, fluency*) as well as aesthetic aspects (*content appeal, composition, color, lighting, and camera trajectory*).

- **WaterlooSQoE-III** (Duanmu et al., 2018): The WaterlooSQoE-III dataset comprises 20 RAW HD reference videos and 450 simulated streaming videos. To generate meaningful and representative test videos, a series of DASH video streaming experiments are conducted, capturing relevant streaming activities and reconstructing the streaming sessions using video processing tools. The WaterlooSQoE-III dataset primarily focuses on assessing the *quality of experience (QoE) in streaming video*.

- **WaterlooSQoE-IV** (Duanmu et al., 2020): The WaterlooSQoE-IV dataset is currently the largest subject-rated VQA dataset for *quality of experience*, featuring 1,350 adaptive streaming videos. These videos are generated from a diverse range of source content, video encoders, network conditions, adaptive bitrate (ABR) algorithms, and viewing devices.

- **T2VQA-DB** (Kou et al., 2024): The T2VQA-DB dataset utilizes 9 different video generation models to create 10,000 AIGC videos. A total of 27 subjects are invited to assess the perceptual quality of each video, focusing on two main aspects: *text-video alignment and video fidelity*. Text-video alignment refers to how well the generated video content corresponds to the given text description, while video fidelity encompasses factors such as distortion, saturation, motion consistency, and content coherence.

- **VideoFeedback** (He et al., 2024b): The VideoFeedback dataset contains human-provided multi-aspect scores for 37.6K synthesized videos generated by 11 different video generative models. The scores assess various aspects, including *visual quality, temporal consistency, dynamic realism, text-to-video alignment, and factual consistency*.

- **LIVE-YT-Gaming** (Yu et al., 2023): The LIVE-YT-Gaming dataset consists of 600 real user-generated gaming videos. A subjective human study is conducted on this dataset, resulting in 18,600 quality ratings provided by 61 participants. The primary focus of the study is on evaluating the *visual quality of the videos.*

We mainly sample the videos from these VQA datasets for the following reasons: 1) The VQA datasets mentioned above feature *well-designed video selection processes and rigorous human annotation standards*. The quality labels can help us control the quality distribution of **Q-Bench-Video**. 2) Moreover, these datasets are mostly focused on quality issues (close to low-level information) and therefore usually *isolated from high-level multi-modal video datasets*. Thus sampling videos from VQA datasets can help prevent overlap with pre-training data used by LMMs to minimize the possibility of data leakage. As a result, these VQA datasets are well-suited to serve as sources for **Q-Bench-Video**, contributing to a more robust and accurate evaluation.

## B  VIDEO SAMPLING APPROACH

We apply a uniform sampling approach directly based on the quality scores of the videos. Specifically, as the **MaxWell** and **VideoFeedback** datasets have multiple labels for one video, we decided to use the *overall quality score* from the **MaxWell** dataset and the *visual quality score* from the **VideoFeedback** dataset as the quality score for the sampling process. For other VQA datasets, where only a single quality score is provided for each video, that score is used for sampling. Given a VQA dataset where each video $v_i$ has an associated quality score $q_i$ in the range $[q_{min}, q_{max}]$, we divide this range into five equal intervals and perform uniform sampling of the scores across these intervals.

**Step 1: Define the Quality Score Range.**

Let $q_i$ represent the quality score for video $v_i$, where the quality score is bounded by:

$$q_{\min} \leq q_i \leq q_{\max}, \quad \forall i, \tag{1}$$

where $q_{\min}$ and $q_{\max}$ represent the min and max quality scores within the VQA dataset.

**Step 2: Divide the Range into Five Equal Intervals.**

To uniformly divide the quality score range $[q_{\min}, q_{\max}]$ into five equal intervals, we first calculate the width of each interval $\Delta q$:

$$\Delta q = \frac{q_{\max} - q_{\min}}{5}, \tag{2}$$

Thus, the five intervals are defined as follows:

$$[q_{\min}, q_{\min} + \Delta q), \quad [q_{\min} + \Delta q, q_{\min} + 2\Delta q), \quad \ldots, \quad [q_{\min} + 4\Delta q, q_{\max}], \tag{3}$$

**Step 3: Uniform Sampling Across the Intervals.**

We can then perform uniform sampling within these intervals. If we want to ensure uniform sampling across the entire score range, the probability of selecting a score from each interval should be the same. Let $X$ be the random variable representing the quality score, and the probability of sampling from any interval $I_j$ is:

$$P(X \in I_j) = \frac{1}{5}, \quad j = 1, 2, 3, 4, 5, \tag{4}$$

where $I_j$ represents the $j$-th interval.

**Step 4: Sampling Within Each Interval.**

Within each interval $I_j = [q_{\min} + (j-1)\Delta q, q_{\min} + j\Delta q)$, a score $X_j$ is sampled uniformly:

$$X_j \sim U\left(q_{\min} + (j-1)\Delta q, q_{\min} + j\Delta q\right), \tag{5}$$

**Final Formula.** The overall uniform sampling process across the five intervals can be described as:

$$X = \begin{cases} U\left(q_{\min}, q_{\min} + \Delta q\right), & \text{with probability } \frac{1}{5}, \\ U\left(q_{\min} + \Delta q, q_{\min} + 2\Delta q\right), & \text{with probability } \frac{1}{5}, \\ \vdots & \\ U\left(q_{\min} + 4\Delta q, q_{\max}\right), & \text{with probability } \frac{1}{5}, \end{cases} \tag{6}$$

This process ensures that the quality scores are uniformly sampled across the entire score range divided into five equal intervals.

## C  ANNOTATION PROCESS

A group of eight experts, all with professional photography skills and extensive experience, participate in the subjective annotation experiment for **Q-Bench-Video**. The experiment takes place in a controlled lab environment with standard indoor lighting. A Dell 4K monitor with a resolution of $3840 \times 2160$ is used to display the interfaces, as shown in Fig 4. To avoid fatigue, each expert labels up to 30 videos per day, and every annotation is carefully reviewed by at least three other experts before final approval. This process ensures the highest level of accuracy and rigor in the **Q-Bench-Video** labels, making performance testing of **Q-Bench-Video** more precise and meaningful.

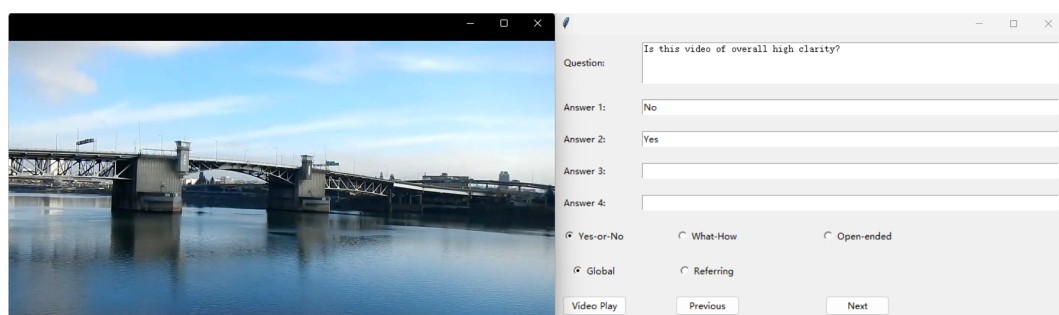

(a) Annotation GUI for single videos of **Q-Bench-Video**.

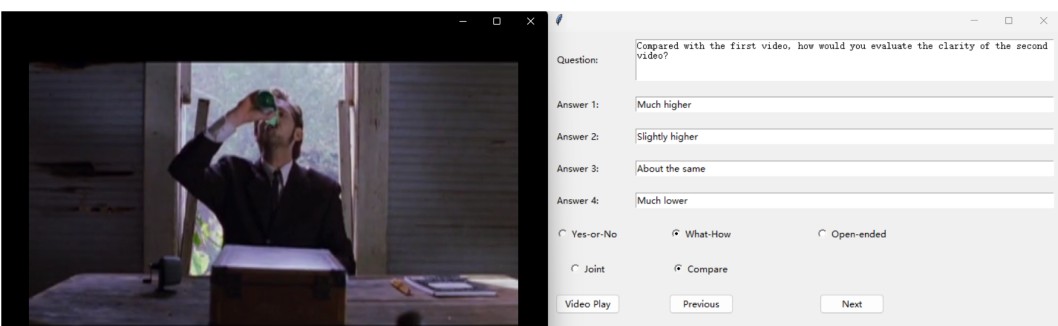

(b) Annotation GUI for video pairs of **Q-Bench-Video**.

Figure 4: Illustration of the annotation GUIs for **Q-Bench-Video**. (a) shows the interface for annotating single videos, where the annotator can select the question type and play the videos using the *Video Play* button. The annotator can also switch to the next and previous annotation with the *Next* and *Previous* buttons. (b) presents the interface for annotating video pairs. When the annotator presses the *Video Play* button, the video pairs are played sequentially, with a five-second gray screen serving as an interval between the two videos.

# D  YES-OR-NO RATIO

As indicated in numerous previous works (Zhang et al., 2024c; Wu et al., 2024b), LMMs often exhibit a bias when answering *Yes-or-No* questions, such as a tendency to favor *Yes* over *No*. To mitigate this issue, we specifically examine the distribution of correct answers for *Yes-or-No* questions and adjust the questions to ensure a balanced 50%/50% ratio between *Yes* and *No* answers. For illustration, we provide an example to demonstrate how we modify the questions:

**Question-answer before modification.**

```
Q: Is this video of high clarity?
A. Yes (Correct) N. No
```

**Question-answer after modification.**

```
Q: Is this video of low clarity?
A. Yes N. No (Correct)
```

# E  BENCHMARK SETTING

## E.1  SAMPLING FRAMES.

Given the sensitivity of temporal quality to the frame number, we ensure fairness in comparisons by standardizing the input to uniformly sample 16 frames. For *Image LMMs*, the frame number is 8. Specifically, for single videos, we sample 16 frames uniformly from each video. For video pairs, we sample 8 frames from each video and create a composite 16-frame input.

### E.2 PROMPT FOR SINGLE VIDEOS ON MCQ

*# User: You will receive [Frame_Num] distinct frames that have been uniformly sampled from a video sequence, arranged in the same temporal order as they appear in the video. Please analyze these frames and answer the question based on your observations. [Question] [Answers] Please answer the question in the following format: the uppercase letter of the correct answer option itself +'.'. Please do not add any other answers beyond this.*

### E.3 PROMPT FOR SINGLE VIDEOS ON OPEN-ENDED QUESTIONS

*# User: You will receive [Frame_Num] distinct frames that have been uniformly sampled from a video sequence, arranged in the same temporal order as they appear in the video. Please analyze these frames and provide a detailed and accurate answer from the perspective of visual quality based on your observations. [Question]*

### E.4 PROMPT FOR VIDEO PAIRS ON MCQ

*# User: You will receive [Frame_Num] distinct frames in total. The first [Frame_Num/2] frames and [Frame_Num/2]-[Frame_Num] frames are uniformly sampled from the first and the second video sequence, arranged in the same temporal order as they appear in the videos. The first video frames: [Frames1]. The second video frames: [Frames2]. Please analyze these frames and answer the questions based on your observations. [Question] [Answers] Please answer the question in the following format: the uppercase letter of the correct answer option itself +'.'. Please do not add any other answers beyond this.*

### E.5 PROMPT FOR VIDEO PAIRS ON OPEN-ENDED QUESTIONS

*# User: You will receive [Frame_Num] distinct frames in total. The [Frame_Num/2] frames and [Frame_Num/2]-[Frame_Num] frames are uniformly sampled from the first and second video sequences, arranged in the same temporal order as they appear in videos. The first video frames: [Frames1]. The second video frames: [Frames2]. Please analyze these frames and provide a detailed and accurate answer based on your observations. [Question]*

## F HUMAN PERFORMANCE ON Q-BENCH-VIDEO

To evaluate the gap between LMMs and human performance in video quality understanding, we invite three human participants to take part in experiments using the `test` subset of **Q-Bench-Video**. The experimental setup and procedure are identical to the annotation environment previously described (See Appendix C). It's worth noting that the participants undergo a brief training session to familiarize themselves with the tasks and acquire the necessary knowledge of video quality. Afterward, they complete the test, and we record their average scores as the final results of human performance. The human performance testing interface is shown in Fig. 5.

## G EVALUATION DETAILS

### G.1 EVALUATION ON MCQs

For MCQ evaluation, we measure the performance directly based on accuracy. However, in cases where LMMs do not directly return an option, we have established the following process: If the LMM returns an option letter as instructed, we directly calculate its accuracy. If the LMM does not respond with an option letter, we use a GPT-involved method (using GPT-4o) to evaluate whether the answer is correct before calculating the accuracy. To mitigate errors due to randomness, we conduct five rounds of testing. An answer is considered correct if it is deemed accurate in three or more of these rounds. The prompt for judging answer correctness is as follows:

*#System: You are a helpful assistant that grades answers related to visual video quality. There are a lot of special terms or keywords related to video processing and photography. You will pay attention to the context of 'quality evaluation' when grading.*

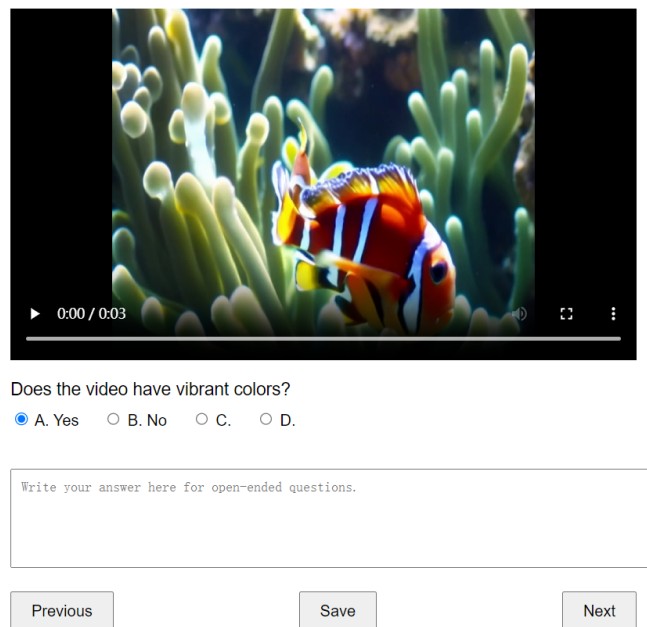

Figure 5: Illustration of the human performance testing interface. The human participants are allowed to select an option as their answer to the MCQ questions or write down their response to the open-ended questions in the text box below.

*#User: You will now be provided with a question [question] and a set of options [answers] with option [correct_answer] being the correct answer. Additionally, there will be an answer [response] provided by a respondent. Please determine whether the respondent's answer is correct considering the context of the question. Even if the word choice is not completely the same, you can decide based on the given options and see whether the one in the answer is close enough to the given correct answer, The result is 1 if the answer is correct and else the result is 0. Please only provide the result in the following format: Score:*

## G.2 MCQ EVALUATION EXAMPLE

*#User: You will now be provided with a question [How is the clarity of the first child that appears in the video? ] and a set of options ["A. Average, with average clarity, some facial details are missing, but movements are smooth", "B. Above average, with good clarity, facial details are relatively clear", "C. Very good, clear frames, with rich facial details and smooth movements", "D. Very poor, with low clarity, facial details are missing, presence of frame drops, and heavy shadowing in movements"] with option ["A. Average, with average clarity, some facial details are missing, but movements are smooth"] being the correct answer. Additionally, there will be an answer ["The first child that appeared in the video had with average clarity, some facial details are missing, but movements are smooth."] provided by a respondent. Please determine whether the respondent's answer is correct considering the context of the question. Even if the word choice is not completely the same, you can decide based on the given options and see whether the one in the answer is close enough to the given correct answer, The result is 1 if the answer is correct and else the result is 0. Please only provide the result in the following format: Score:*

***5-round GPT score: [1, 1, 1, 1, 1]***

## G.3 EVALUATION ON OPEN-ENDED QUESTIONS

For evaluating open-ended questions, we also employ a 5-round GPT-involved evaluation strategy. GPT (using GPT-4o) is tasked with scoring the LMM responses based on three criteria: completeness, accuracy, and relevance, with scores from $\{0, 1, 2\}$. To facilitate calculation, we sum the

Q: How is the clarity of the first child that appears in the video?
A. Average, with average clarity, some facial details are missing, but movements are smooth ✓
B. Above average, with good clarity, facial details are relatively clear
C. Very good, clear frames, with rich facial details and smooth movements
D. Very poor, with low clarity, facial details are missing, presence of frame drops, and heavy shadowing in movements

✗ **LLaVA-Next**: C
✓ **LLaVA-v1.5**: A
✗ **mPLUG-Owl2**: C
✗ **mPLUG-Owl3**: D.
✗ **LLaVA-OneVision**: C
✗ **InternVL-Chat**: D
✓ **VILA1.5**: A.
✗ **PLLaVA**: D
✓ **ST-LLM**: The first child that appeared in the video had with average clarity, some facial details are missing, but movements are smooth.

✗ **LLaVA-Next-Video**: D
✓ **Video-LLaVA**: A
✓ **VideoChat2**: A. with average clarity, some facial details are missing, but movements are smooth 
✗ **Gemini 1.5 Flash**: C
✓ **Gemini 1.5 Pro**: A.
✗ **GPT-4o mini**: C.
✗ **GPT-4o**: C.
✓ **GPT-4 Turbo**: A.
✓ **Human**: A

Figure 6: Qualitative comparison for LMM on MCQ response.

scores from all five rounds and normalize the total to a range between 0 and 1, which is then used as the 'accuracy' performance. Consider the evaluation of open-ended questions using a 5-round GPT-involved scoring system. Let $S_i$ be the score assigned by GPT in the $i$-th round, which assesses the response based on three criteria: completeness, accuracy, and relevance. Each score $S_i$ can take a value in $\{0, 1, 2\}$. The total score accumulated over the five rounds is calculated as:

$$T = \sum_{i=1}^{5} S_i$$

To normalize this score into an accuracy metric between 0 and 1, we use the following normalization:

$$\text{Accuracy} = \frac{T}{10}$$

This normalization assumes the maximum possible score $T$ is 10, corresponding to a perfect score of 2 across all 5 rounds. The prompt for evaluation on open-ended questions is as follows:

*#System: You are a helpful assistant that grades answers related to visual video quality. There are a lot of special terms or keywords related to video processing and photography. You will pay attention to the context of 'quality evaluation' when grading.*

*#User: Given the question [question], evaluate whether the response [response] completely matches the correct answer [correct_answer]. First, check the response and please rate score 0 if the response is not a valid answer. Please rate score 2 if the response completely or almost completely matches the correct answer on completeness, accuracy, and relevance. Please rate score 1 if the response partly matches the correct answer on completeness, accuracy, and relevance. Please rate score 0 if the response doesn't match the correct answer on completeness, accuracy, and relevance at all. Please only provide the result in the following format: Score:*

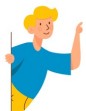

**Question:** The little girl in the lower part of this video appears with good visual quality, please briefly analyze the reasons.
**Annotation Response:** The overall lighting of the video is appropriate and even. The contrast and saturation are natural, and the camera is stable without shaking. This makes the little girl's facial features, hair, and overall movements appear clear and natural, giving a high visual quality presentation.

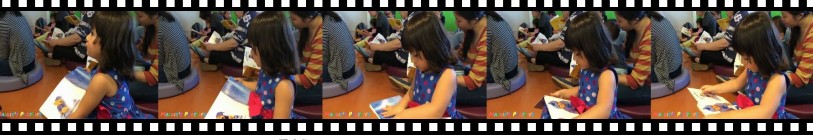

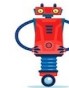
**LLaVA-Next-Video:** The video shows a young girl sitting and reading a book. The quality of the video is good because it is a close-up shot, which allows for clear visibility of the girl's facial features and the details of the book she is reading. The lighting is also good, as it is bright enough to see the text on the book clearly. The focus is sharp, capturing the girl's expression and the book's details without any distortion. The composition of the video is also well-balanced, with the girl centrally positioned and the book in focus.

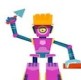
**VILA1.5:** The little girl in the lower part of the video screen appears with good quality because the video is shot in high resolution.

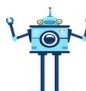
**PLLaVA:** The little girl in the lower part of the video screen appears with good quality because she is the main subject, and her actions and expressions are the focus of the viewer's attention. The camera is positioned to capture her face and upper body, which are the most expressive parts of a person. Additionally, the lighting and composition of the video emphasize her features, making her appear more prominent. The other individuals in the video are not as clearly visible or detailed, which helps to draw the viewer's attention to the girl.

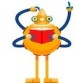
**InternVL-Chat:** The little girl in the lower part of the video screen appears with good quality because she is the main subject of the video. The focus is on her actions, such as reading a book, which is the primary activity in the scene. The other people in the background are not the main focus, so their video quality is not as high as the little girl's.

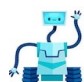
**LLaVA-OneVision:** The little girl in the lower part of this video screen appears with good quality due to several factors. Firstly, the lighting in the room is bright and even, which helps to illuminate her face and clothing clearly without harsh shadows. Secondly, the camera's focus is sharp on her, allowing for the details of her dress, hair, and facial features to be visible. Lastly, the resolution of the video seems high, contributing to the clarity and detail that can be observed in her appearance.

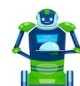
**mPLUG-Owl3:.** The little girl in the video appears with good quality due to several factors. Firstly, the lighting in the room is even, which helps to ensure that the details of her face and clothing are clearly visible. Secondly, the focus on the girl is sharp, allowing for a clear view of her expressions and the book she is reading. Lastly, the color saturation and contrast are well-balanced, making the colors of her dress and the book stand out without being oversaturated.

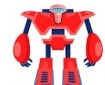
**Gemini 1.5 Pro:** The little girl is in focus while the background is slightly out of focus, indicating a shallow depth of field. This technique helps to draw attention to the main subject, the little girl, by blurring the less important background. This is a common practice in videography and photography to create visually appealing and engaging content.

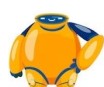
**GPT-4o:** The video frames depict a sequence where a little girl is sitting and reading a book in what appears to be a reading class or group setting. Here are some observations regarding the quality and content of the frames featuring the girl:
1.Stable Scene: The girl remains relatively stationary throughout the sequence, focused on her book. This lack of movement helps maintain a clear and sharp video without motion blur.
2.Good Lighting: The lighting appears consistent and sufficient, contributing to the overall high quality of the video. Proper lighting can significantly affect the clarity and perceived quality of a video frame
3.Proximity and Focus: The girl is prominently in the foreground and seems to be the focal point of the camera. The close proximity and camera focus on her enhance the detail and sharpness of her video.
4.Color Contrast: The girl is wearing a brightly colored dress with a distinct pattern. This contrast with the background and other elements in the room makes her stand out, adding to the perceptual quality of her video.
5.Composition: The framing is well-composed, with the girl positioned centrally or prominently in the frame. This effective composition helps draw viewer attention to her, ensuring that she is a clear and central figure in the video sequence.
6.Quality of Recording Equipment: Assuming the video was recorded with a good-quality camera, the equipment itself would contribute to achieving a higher resolution and better quality frames.
Given these factors, the overall quality of the frames featuring the girl can be attributed to good lighting, stability, focus, composition, and possibly high-quality recording equipment.

Figure 7: Qualitative comparison for LMM on ppen-ended question response.

Table 4: Results on the `dev` subset for the video quality perception ability of LMMs.

| Sub-categories | Question Types | | | Quality Concerns | | | | Overall↑ |
|---|---|---|---|---|---|---|---|---|
| **LMM** (*LLM*) | *Yes-or -No↑* | *What -How↑* | *Open -ended↑* | *Tech.↑* | *Aes.↑* | *Temp.↑* | *AIGC↑* | |
| **Random guess** | 50.00% | 25.00% | 0.00% | 25.74% | 21.98% | 26.56% | 25.54% | 27.14% |
| *Open-source Image LMMs* | | | | | | | | |
| LLaVA-Next (*Mistral-7B*) | 63.20% | 43.78% | 30.42% | 45.95% | 54.83% | 45.63% | 46.24% | 47.00% |
| LLaVA-v1.5 (*Vicuna-v1.5-13B*) | 53.40% | 46.87% | 33.85% | **55.83%** | 55.90% | 44.91% | 45.96% | 45.57% |
| mPLUG-Owl2 (*LLaMA2-7B*) | 59.61% | 38.83% | 31.57% | 42.49% | 53.28% | 44.73% | 40.07% | 44.20% |
| *Open-source Video LMMs* | | | | | | | | |
| mPLUG-Owl3 (*Qwen2-7B*) | 60.82% | **56.52%** | 35.84% | 51.34% | 60.46% | **54.26%** | 37.30% | **52.44%** |
| LLaVA-OneVision (*Qwen2-7B*) | 62.13% | 52.23% | **38.56%** | 48.74% | **61.53%** | 48.81% | 44.57% | 52.12% |
| InternVL-Chat (*Vicuna-7B*) | **70.21%** | 48.65% | 32.20% | 50.24% | 49.50% | 52.96% | 47.69% | 51.91% |
| VILA1.5 (*LLaMA3-8B*) | 61.59% | 47.30% | 36.88% | 46.74% | 59.30% | 47.57% | 43.67% | 49.62% |
| PLLaVA (*Mistral-7B*) | 65.13% | 54.23% | 29.44% | 50.31% | 60.09% | 50.13% | **50.75%** | 51.23% |
| LLaVA-Next-Video (*Mistral-7B*) | 65.98% | 45.31% | 31.92% | 48.11% | 57.33% | 47.09% | 45.56% | 48.97% |
| ST-LLM (*Vicuna-v1.1-7B*) | 46.43% | 28.45% | 32.31% | 33.32% | 45.66% | 36.01% | 32.62% | 35.89% |
| Video-LLaVA (*Vicuna-v1.5-7B*) | 64.36% | 39.38% | 30.86% | 43.35% | 55.97% | 45.58% | 43.64% | 45.89% |
| VideoChat2 (*Mistral-7B*) | 56.54% | 33.13% | 35.36% | 39.09% | 49.27% | 41.59% | 38.04% | 42.06% |
| *Proprietary LMMs* | | | | | | | | |
| **Gemini 1.5 Flash** | 66.21% | 59.36% | 46.06% | 54.01% | 67.31% | 56.89% | 52.23% | 57.99% |
| **Gemini 1.5 Pro** | 65.93% | **61.33%** | **47.15%** | **56.23%** | **68.23%** | 56.00% | **54.04%** | **58.36%** |
| **GPT-4o mini** | 63.00% | 50.65% | 42.16% | 49.66% | 60.88% | 50.31% | 43.84% | 52.78% |
| **GPT-4o** | **67.21%** | **58.36%** | 45.50% | 54.15% | 67.50% | **57.08%** | 52.39% | 58.11% |
| **GPT-4 Turbo** | 66.67% | 58.46% | 44.18% | 54.53% | 63.99% | 51.96% | 45.73% | 56.40% |

## G.4 Open-ended Question Evaluation Example

*#User: Given the question [“The little girl in the lower part of this video appears with good visual quality, please briefly analyze the reasons.”], evaluate whether the response [“The little girl in the lower part of the video screen appears with good quality because the video is shot in high resolution.”] completely matches the correct answer [“The overall lighting of the video is appropriate and even. The contrast and saturation are natural, and the camera is stable without shaking. This makes the little girl’s facial features, hair, and overall movements appear clear and natural, giving a high visual quality presentation.”]. First, check the response and please rate score 0 if the response is not a valid answer. Please rate score 2 if the response completely or almost completely matches the correct answer on completeness, accuracy, and relevance. Please rate score 1 if the response partly matches the correct answer on completeness, accuracy, and relevance. Please rate score 0 if the response doesn’t match the correct answer on completeness, accuracy, and relevance at all. Please only provide the result in the following format: Score:*

***5-round GPT score: [0, 0, 1, 0, 0] Final Score: 1/10 = 0.1***

## H Qualitative Visualization Results

In Fig. 6, we present an example of LMM responses to MCQ questions. First, it is clear that most LMMs can follow the instructions well to select the option they believe is correct, except for ST-LLM, which requires further checking to determine if the answer is correct (See Appendix G for details about GPT-assisted evaluation approach). Secondly, for basic quality assessments like clarity, which are relatively simple for humans, about half of the LMMs still failed to answer correctly. This again highlights the need to improve LMM video understanding capabilities.

In Fig. 7, we further highlight a case involving the top-performing LMM responding to the open-ended question. In contrast to the relatively straightforward multiple-choice questions, there is a noticeable performance gap among LMMs when addressing open-ended questions. For example, GPT-4o delivers the most detailed and accurate responses, while VILA1.5 produces significantly shorter and less comprehensive answers. This variation in performance on open-ended tasks underscores the current instability in LMMs’ video quality understanding.

Table 5: Results on the `dev` subset for the video quality perception ability across single videos and video pairs of LMMs.

| Sub-categories | Single Videos | | | Video Pairs | | | |
| --- | --- | --- | --- | --- | --- | --- | --- |
| **LMM** *(LLM)* | *Global*↑ | *Referring*↑ | *Overall*↑ | *Joint*↑ | *Compare -fine*↑ | *Compare -coarse*↑ | *Overall*↑ |
| **Random guess** | 22.29% | 29.15% | 25.67% | 31.47% | 32.45% | 30.80% | 31.50% |
| *Open-source Image LMMs* | | | | | | | |
| LLaVA-Next (*Mistral-7B*) | **62.83%** | 45.14% | 33.69% | 46.38% | 57.86% | 47.84% | 48.46% |
| LLaVA-v1.5 (*Vicuna-v1.5-13B*) | 45.91% | **55.01%** | 50.42% | 33.41% | 39.11% | 33.37% | 35.39% |
| mPLUG-Owl2 (*LLaMA2-7B*) | 45.80% | 45.05% | 45.43% | 55.14% | 46.00% | 38.12% | 44.14% |
| *Open-source Video LMMs* | | | | | | | |
| mPLUG-Owl3 (*Qwen2-7B*) | 51.71% | 48.78% | 50.26% | 56.60% | 51.52% | 65.75% | 59.03% |
| LLaVA-OneVision (*Qwen2-7B*) | 49.41% | 49.35% | 49.38% | **57.93%** | **64.53%** | **66.98%** | **64.39%** |
| InternVL-Chat (*Vicuna-7B*) | 49.43% | 54.05% | **51.72%** | 44.74% | 53.67% | 52.14% | 50.50% |
| VILA1.5 (*LLaMA3-8B*) | 46.55% | 48.69% | 47.61% | 54.17% | 48.25% | 53.10% | 51.61% |
| PLLaVA (*Mistral-7B*) | 48.21% | **55.01%** | 51.58% | 35.69% | 57.45% | 52.16% | 50.86% |
| LLaVA-Next-Video (*Mistral-7B*) | 47.01% | 52.23% | 49.59% | 29.83% | 52.74% | 51.22% | 47.66% |
| ST-LLM (*Vicuna-v1.1-7B*) | 34.06% | 37.93% | 35.98% | 24.83% | 35.47% | 39.71% | 35.38% |
| Video-LLaVA (*Vicuna-v1.5-7B*) | 43.66% | 47.73% | 45.67% | 41.86% | 44.30% | 58.91% | 50.54% |
| VideoChat2 (*Mistral-7B*) | 38.91% | 40.47% | 39.68% | 50.52% | 48.11% | 50.07% | 49.47% |
| *Proprietary LMMs* | | | | | | | |
| **Gemini 1.5 Flash** | 53.83% | 56.25% | 55.03% | 47.48% | 67.83% | 69.09% | 64.03% |
| **Gemini 1.5 Pro** | 51.98% | **60.80%** | **56.35%** | 43.41% | **68.30%** | 69.81% | 64.13% |
| **GPT-4o mini** | 50.03% | 49.88% | 49.95% | 46.69% | 60.53% | 66.39% | 59.36% |
| **GPT-4o** | **57.98%** | 54.39% | 56.17% | 48.00% | 67.64% | **70.17%** | **65.09%** |
| **GPT-4 Turbo** | 55.03% | 55.47% | 55.25% | **49.76%** | 62.60% | 64.29% | 60.81% |

---

**Algorithm 1** Classification of Video Pairs in VQA Dataset

1: **Input:** Set of videos $V$ with quality scores
2: **Output:** Classifications of video pairs as Compare-coarse or Compare-fine
3: Initialize list of video pairs $P$ to empty
4: **for** each pair $(v_i, v_j) \in V \times V$, $i \neq j$ **do**
5:     Add $(v_i, v_j)$ to $P$
6: **end for**
7: **Randomly select** pairs from $P$ without repetition
8: Calculate $\Delta q_{ij} = |q_i - q_j|$ for each $(v_i, v_j) \in P$
9: **Rank** all pairs $(v_i, v_j)$ by $\Delta q_{ij}$
10: $\theta \leftarrow$ **Median** of all $\Delta q_{ij}$
11: **for** each $(v_i, v_j) \in P$ **do**
12:     **if** $\Delta q_{ij} > \theta$ **then**
13:         Label $(v_i, v_j)$ as *Compare-coarse*
14:     **else**
15:         Label $(v_i, v_j)$ as *Compare-fine*
16:     **end if**
17: **end for**

---

## I  PERFORMANCE ON THE `dev` SUBSET

The performance results on the `dev` subset of **Q-Bench-Video** are illustrated in Table 4 and Table 5. This subset is planned to be opened to the public in the future. As such, the performance results will serve primarily as a reference. Currently, all evaluated LMMs have not been exposed to this subset, making it suitable for cross-validation with the `test` subset. Although there are slight differences in LMM performance between the `dev` and `test` subsets, the overall gap is minimal, essentially maintaining the performance trends and rankings of the LMMs. Specifically, LLaVA-OneVision and mPLUG-Owl3 continue to hold the top two spots among open-source models, while GPT-4o and Gemini 1.5 Pro lead among proprietary models, suggesting that **Q-Bench-Video** is a reliable and stable benchmark for video quality understanding.

## J    COMPARISON FOR VIDEO PAIRS

In this section, we focus on discussing how to categorize the annotations into Compare-fine and Compare-coarse classifications. We collect videos from the VQA dataset that already have annotated quality scores. Since our comparisons are confined to video pairs from the same VQA dataset source, the quality scores between video pairs are valid and meaningful. Within single VQA dataset, we randomly select video pairs without repetition, and then rank all video pairs based on the differences in their quality scores. A median value is then chosen as the threshold. Pairs with a difference exceeding this threshold are labeled as Compare-coarse, while those with a difference below it are labeled as Compare-fine. The pseudocode for this procedure is detailed in Algorithm 1.

## K    LIMITATIONS & SOCIAL IMPACT

**Limitations. 1**) *Subjectivity in Evaluation*: Although the benchmark includes efforts to minimize subjective bias by using expert annotations, aesthetic aspects such as visual appeal and composition inherently involve subjective judgments. Even among trained experts, there might be variations in opinions on what constitutes high or low aesthetic quality. **2**) *Rapid Evolution of AIGC Distortions*: The benchmark includes evaluation specifically tailored to AIGC distortions. However, given the fast-paced advancements in AI-generated video technology, future generations of generative models may produce fewer visible distortions or entirely new types of artifacts. This implies that the current version of **Q-Bench-Video** might partly become outdated in the future.

**Social Impact.** By focusing on video quality understanding, this benchmark encourages the development of LMMs that can discern not only the semantic content but also the technical and aesthetic quality of videos. This has broad applications, from improving video compression algorithms to enhancing user experience in media platforms. Ultimately, **Q-Bench-Video** could lead to the creation of better tools for optimizing video quality across diverse industries.

