# OpenReview forum: "Q-Bench-Video: Benchmarking the Video Quality Understanding of LMMs"
_ICLR.cc/2025/Conference — ICLR 2025 Conference Withdrawn Submission_

### Official Review · Reviewer_xK9V · 2024-11-03

**Soundness:** 2
**Presentation:** 2
**Contribution:** 2
**Rating:** 3
**Confidence:** 4

**Summary:**

The paper highlights the shortcomings of existing LMM benchmarks in evaluating perceptual video quality discernibility. To address this limitation, the paper proposes a video quality-specific benchmark called Q-Bench-Video composed of real, AI-generated, and computer graphics video content. To test the LMMs, the paper collects a series of Yes-No, What-How, and Open-ended question-answer pairs about the videos. Video-pair comparisons are also considered. A set of 17 LMMs (12 open-source + 5 proprietary) are benchmarked on Q-Bench-Video. The performance of these LMMs is compared with human performance. The comparison leads to the observation that humans > LMMs > random choice in terms of accuracy.

**Strengths:**

1. The problem is motivated well, and the solution is timely given the ever-increasing usage of LMMs in VQA.

2. The choice of real, AIG, and CG videos, along with the QA pairs, is well thought out.

3. The coverage of the LMMs is extensive.

**Weaknesses:**

1. In my opinion, the work’s full impact is not felt in the current form of the paper. Specifically, the key observation of humans > LMMs > random choice is neither surprising nor unexpected. The same can be said of the observation that proprietary models outperform open-source models. In light of this, the study should have been extended to show how the benchmark can help improve LMM performance to reduce the gap with human performance. Without this, a practitioner would be left wanting for the utility of Q-Bench-Video.

2. Given that the original content sources contained a much larger set of videos, the size of Q-Bench-Video (1800) is too small to qualify as a representative benchmark. For example, the LSVQ dataset has 39000 videos, and the MaxWell dataset has 4500 videos. Further, several user-generated content datasets, such as YT-UGC, Konvik-1K, etc., have in excess of 1500 videos. Given that Q-Bench-Video is proposed as a benchmark composed of not only real videos but also AIG and CG videos, a representative dataset should contain at least 4500 videos.

3. Some of the key details of the work have been relegated to the appendices. For example, the video selection and evaluation strategy are both detailed in the appendix.

4. The fact that only three human subjects were used for human performance evaluation does not inspire confidence in the work. There is no information on statistical consistency tests on the subjective experiments.

5. The presentation quality needs to be improved.

**Questions:**

1. Can you please offer more useful insights into your findings? The main result of human > LMM > random and the other observations in Section 4 are not unexpected. For example, let's consider the following observation - "This disparity underscores a significant proficiency gap between LMMs’ capability in handling straightforward, closed-form questions and their effectiveness in navigating the complexities of real-world problem-solving, particularly in the context of video quality evaluation." Can you please provide any reasons for such a gap?

2. How can Q-Bench-Video be used to improve LMM performance? Please provide some guidance on this aspect.

3. What factors led to the chosen dataset size?

4. What is the frame rate and the duration of the videos? What is the impact of sampling these videos uniformly?

5. Why were only three subjects used for the human evaluation? Is this a statistically significant number? Please refer to https://www.itu.int/dms_pubrec/itu-r/rec/bt/R-REC-BT.500-15-202305-I!!PDF-E.pdf. Section 2.5.1 explicitly mentions 15 observers for a subjective evaluation.

6. Were statistical consistency tests conducted on the human evaluations? Please refer to Annex 1 of the above ITU document.

7. Can you please clarify the dev and test subsets?

---

### Official Review · Reviewer_BVGh · 2024-11-04

**Soundness:** 2
**Presentation:** 3
**Contribution:** 2
**Rating:** 5
**Confidence:** 4

**Summary:**

This paper presents a novel video benchmark aimed at evaluating the capabilities of large multi-modal models (LMMs) in understanding and assessing video quality. The benchmark stands out due to its inclusion of a wide range of videos sourced from diverse platforms, with a particular emphasis on AI-generated content (AIGC) and CG videos. The authors strategically categorize evaluation questions into three distinct groups to enable a thorough analysis of video quality. Notably, the inclusion of open-ended questions allows for the assessment of complex, nuanced scenarios beyond simple objective measures. Last but not least, the authors conducted extensive comparisons of current LLMs on this benchmark, showing their limited performance on evaluating video quality. Despite these contributions, there are concerns that need to be further addressed before this paper can be accepted.

**Strengths:**

- The paper is well-presented, with adequate examples and figures to show the query designs in the benchmark.

- The questions can cover extensive aspects of video quality, including technical, aesthetic, temporal, and AIGC distortions.

- The video pairs comparison is a good task for evaluating LLM's capability, which can be considered as an advantage over existing video benchmarks.

**Weaknesses:**

- Motivation of the Benchmark: The paper's motivation for building the benchmark is not sufficiently strong. While the introduction states the importance of video quality for viewer experience and high-quality video generation, what viewers primarily care about is the classification of video quality (e.g., low, medium, high). A well-trained classifier could fulfill this need more directly. Additionally, for assessing video generation quality, designing robust evaluation metrics is more crucial than simply building a benchmark to test LLMs.

- Question Design Limitations: The paper’s three question categories are not comprehensive enough. It would be beneficial to include overarching questions such as "What is the quality level of this video?" with clear, concise answers like "low/medium/high." This would better align with practical applications and user expectations.

- Evaluation Metric Weaknesses: The evaluation metrics used to assess LLMs' QA performance lack robustness. The current scoring prompt only accounts for completeness, accuracy, and relevance, which are insufficient for open-ended questions. Recent studies (e.g., RAGchecker [NeurIPS 2024]) suggest including metrics like faithfulness (to detect hallucinations), truthfulness, correctness, etc. Additionally, prompts should be specifically designed with definitions and in-context examples to help LLMs understand each metric. Without refining the prompt and using comprehensive evaluation metrics, the reported performance scores are likely unreliable.

**Questions:**

- How to generate questions for different categories? If you use LLM to generate the questions, please include the prompt or template in the paper for readers to study. If not, illustrate your approach with details.

---

### Official Review · Reviewer_Wt97 · 2024-11-04

**Soundness:** 3
**Presentation:** 4
**Contribution:** 3
**Rating:** 6
**Confidence:** 4

**Summary:**

The paper presents Q-Bench-Video, a comprehensive benchmark designed to assess the video quality understanding capabilities of Large Multi-modal Models (LMMs). It focuses specifically on video quality, incorporating various video types, including natural, AI-generated content (AIGC), and computer graphics (CG). The benchmark evaluates models across multiple question types (Yes-or-No, What-How, and Open-ended) and quality concerns (Technical, Aesthetic, Temporal, and AIGC distortions), providing a new assessment framework. Results show that there is still a notable gap between LMM and human performance.

**Strengths:**

1. The benchmark includes a broad range of video types, including natural scenes, AIGC, and CG, enhancing the evaluation’s comprehensiveness.
2. By integrating various question types and quality concerns, It offers a novel framework for assessing different aspects of video quality.
3. The benchmark includes 2,378 question-answer pairs curated by experts, providing a reliable evaluation dataset.

**Weaknesses:**

1. Although the benchmark includes annotations reviewed by three additional participants, the performance on human evaluation is only 81.56%. This raises concerns about whether the annotations meet a robust confidence interval and whether there may be inherent biases in the data. The reliance on subjective assessments for video quality (such as aesthetics) could introduce inconsistencies.
2. The technical and aesthetic evaluations in the dataset are annotated by eight experts, but the paper does not validate these annotations against established subjective scores, such as those from the MaxWell dataset.
3. The paper provides no details on the video format and resolution used in the benchmark. It is unclear if any downsampling was applied, which may affect the quality assessment and limit the applicability of the results to different resolutions.
4. The benchmark’s evaluation of open-ended questions relies on GPT-based scoring, repeated five times for consistency. However, this method may introduce biases, as GPT may favor responses with longer outputs or struggle with certain scenarios where perfect answers are difficult to generate. This raises concerns about the objectivity and reliability of the scoring for open-ended questions.
5. The paper does not provide a comparison with other established video quality datasets to validate the reliability of its annotations.

**Questions:**

1. The phrasing in line 465 is unclear, potentially causing confusion for readers. Could you please explain it more.
2. In the video pair comparisons, it is unclear if the benchmark includes comparisons of the same video at different quality levels or comparisons across entirely different videos with varying quality levels.
3. Could you please provide the selected video information about the formats and resolution and so on.

---

### Official Review · Reviewer_W4DH · 2024-11-04

**Soundness:** 3
**Presentation:** 3
**Contribution:** 3
**Rating:** 5
**Confidence:** 5

**Summary:**

This paper presents Q-Bench-Video, a benchmark specifically designed to evaluate the video quality understanding capabilities of Large Multi-modal Models (LMMs). Recognizing a gap in existing benchmarks that focus on high-level video comprehension rather than quality assessment, Q-Bench-Video targets four key dimensions of video quality: technical, aesthetic, temporal, and AI-generated content (AIGC) distortions. The benchmark includes a diverse set of videos from natural scenes, AI-generated content, and computer graphics, ensuring a balanced distribution across quality levels. To comprehensively assess LMMs, Q-Bench-Video employs various question types—Yes-or-No, What-How, open-ended, and video pair comparisons—that capture the models’ performance across straightforward and complex tasks. Validated on 17 LMMs (12 open-source and 5 proprietary), the benchmark reveals that, while LMMs demonstrate foundational capabilities in video quality assessment, they fall significantly short of human-level performance, particularly in open-ended and AIGC-specific queries. Q-Bench-Video provides a new standard for evaluating video quality understanding and aims to drive future research on enhancing LMMs’ video quality perception.

**Strengths:**

1. Q-Bench-Video is the first benchmark specifically focused on assessing video quality understanding in Large Multi-modal Models (LMMs), addressing a unique and underexplored aspect of LMMs that goes beyond typical video comprehension. By evaluating quality-related distortions—including technical, aesthetic, temporal, and AIGC-specific aspects—it provides a novel and comprehensive perspective on video quality assessment.

2. The benchmark is meticulously designed with diverse video sources (natural, AIGC, CG) and a balanced quality distribution through uniform sampling from quality-annotated datasets. The use of various question types (Yes-or-No, What-How, open-ended, and video pair comparisons) creates a robust framework that evaluates LMMs across both straightforward and complex scenarios, ensuring a thorough assessment of model capabilities.

3. Q-Bench-Video addresses an important gap in LMM evaluation by focusing on video quality understanding, which is essential for applications in video generation, content moderation, and quality control. The findings—highlighting substantial performance gaps between LMMs and human evaluators, especially in AIGC distortions—offer valuable insights for advancing LMMs and guide future research on improving their quality perception.

**Weaknesses:**

1. While Q-Bench-Video includes diverse video types, there is limited exploration of how LMMs generalize across these domains. Testing LMMs on specific domains, such as medical or surveillance videos, would enhance the benchmark’s relevance by showing how well models handle domain-specific quality variations, which are critical in many real-world applications.

2. The benchmark results indicate that LMMs struggle significantly with open-ended questions, but the paper lacks a detailed breakdown of common error types or patterns in these responses. A more granular analysis could provide clearer insights into where models fail in nuanced video quality understanding and offer guidance on specific areas for model improvement.

3. The reliance on GPT-assisted evaluation for scoring open-ended responses may introduce subjectivity or bias, as the evaluation depends on the alignment of the assistant model’s judgments with human interpretation. Including human evaluations as a baseline for open-ended questions would provide a stronger reference and help validate the reliability of the GPT-assisted scores.

4. The benchmark primarily focuses on a balanced quality distribution but lacks an emphasis on evaluating LMMs in scenarios with challenging low-quality or noisy data, which are common in real-world settings. Incorporating more degraded video samples could better test the robustness of LMMs and highlight areas where models may require improvement for real-world deployment.

5. Some of the highly relevant video quality assessment works should be mentioned in the related work section, like [R1][R2][R3].

[R1] UGC-VQA: Benchmarking Blind Video Quality Assessment for User Generated Content, TIP 2021
[R2] RAPIQUE: Rapid and accurate video quality prediction of user generated content, OJSP 2021
[R3] FAVER: Blind Quality Prediction of Variable Frame Rate Videos, SPIC 202

**Questions:**

1. Could you provide more detailed insights into the specific error patterns observed in LMMs’ open-ended responses?

2. Have you considered testing LMMs on additional specialized domains, such as gaming videos, animation, or screen content, to better assess cross-domain generalization?

3. Given the potential bias in GPT-assisted evaluation, did you consider incorporating human evaluations as a baseline for open-ended responses?

4. Would you consider adding a more challenging subset focused specifically on low-quality or noisy videos?

---

### Official Review · Reviewer_4a37 · 2024-11-04

**Soundness:** 3
**Presentation:** 3
**Contribution:** 3
**Rating:** 5
**Confidence:** 4

**Summary:**

The authors propose a new benchmark for assessing the ability of large multimodal models (LMMs) to understand video quality. They describe the methodology for compiling the benchmark from various datasets. Quantitative and qualitative results of the Video Quality Understanding capability of LMMs are presented based on their benchmark.

**Strengths:**

The methodology for compiling the dataset is described in detail. The experiments are quite extensive: 17 VLM models and several datasets, including AIGC.

**Weaknesses:**

Testing only small-sized LMM models. There is no indication of how the scaling law affects the model's ability to understand video quality.

The length of the VLM answer is not taken into account. Thus, if the model gave a long answer, it can receive higher score.

Technical details regarding the operation of LMM are lacking:
* Details on how exactly videos were input into each LMM are not thoroughly described, such as resolution and the number of frames (see the questions section).

**Questions:**

The MOS ratings in AIGC datasets encompass not only the immediate quality of the video but also other aspects, such as text-to-video alignment. How did you ensure that the samples selected from the T2VQA-DB dataset did not contain errors in text-to-video alignment, so that the MOS could be regarded as an evaluation of quality alone?

You state, "However, in many real-world scenarios, Open-ended Questions, which do not restrict responses to a predefined set, are often more necessary and challenging for LMMs... By adopting this form of questioning, we can better assess an LMM’s ability to perceive video quality in real-world conditions." Could you provide examples of real-world scenarios where models indeed require responses to Open-ended Questions specifically in the context of video quality assessment?

It is mentioned that for Video-LMM, 16 frames were sampled from each video, and for Image-LMM, 8 frames were sampled unless otherwise specified. Details are not disclosed; for instance, the authors of Video-LLaVA state in their paper that the model only accepts 8 frames. In the paper, two videos with 8 frames each (as authors say) are submitted in some tests. Since no other information is provided, it seems either the details of how the videos were input into each model are missing, or the authors may have incorrectly submitted the videos to certain models. Could you please clarify this concern?

---

### Note · Authors · 2024-11-13

I have read and agree with the venue's withdrawal policy on behalf of myself and my co-authors.